# EXPLAINABLE MEDICAL IMAGE CLUSTERING

## ABSTRACT

Image classification is a pivotal task within computer vision, entailing the assignment of labels to entire images. Nonetheless, the complete supervision necessary for such microwell image classification demands extensive annotations, a process that can prove time-intensive to accomplish. Furthermore, situations arise where delving into the intrinsic attributes of data is desired, even when data labels remain uncertain. In this paper, we introduce a cell dataset that captures the developmental trend of cancer cells and T Cells under the influence of diverse experimental conditions medications. Concurrently, we present an approach to both cluster input images and elucidate the rationale behind their grouping. We leverage a U-net encoder for individual microwell image information encoding and a multi-head attention layer for information encapsulation across different time points to achieve this. After clustering, we employ various techniques to expound upon our clustering outcomes. Specifically, we utilize Grad-CAM for visual explication, coupled with human-friendly textual generation to facilitate comprehension of trends within each cluster. Our study encompasses a comparison of diverse architectural models on our proposed dataset, conclusively demonstrating the superior performance of its architecture. Experimental analyses and ablation studies further substantiate the advantages conferred by our innovative architecture.

## 1 INTRODUCTION

Chimeric antigen receptor T cell (CAR-T) therapy has become a significant focus in immunotherapy research because of its potential to treat various diseases. One of its crucial aspects is the integration of several components, including the receptor, spacer/transmembrane, and intracellular signaling domains. Interestingly, the affinity of the binding domain, such as the scFv or VH domain, doesn't always correlate with CAR-T's effectiveness (Sterner & Sterner (2021)). Minor modifications in the design can lead to unexpected changes in toxicity and behavior of CAR-T cells (Roex et al. (2020)). Despite their importance in evaluating CAR-T designs, traditional high-throughput screenings come with limitations. They can be costly and time-consuming, and producing enough purified CAR-T cells for thorough testing is challenging. This often restricts evaluations to a limited number of candidates (Maus & Levine (2016) Shank et al. (2017)). Additionally, most commercial single-cell assays focus primarily on short-term T-cell behaviors, which might not provide a comprehensive view of their function over longer periods. Given the limitations of current methods, there's a growing interest in incorporating image-based deep learning analysis. Deep learning offers a way to process large data sets and could help understand the nuances of T cell/tumor co-culture interactions. This study introduces a new open-culture co-culture assay system, Trovo, designed to work alongside deep learning. Please refer to the appendix for more information on our system.

Despite the notable advancements achieved thus far, using deep learning in medical imaging still grapples with certain limitations. Given the unique characteristics intrinsic to medical images, both patients and medical professionals have expressed a compelling need for comprehensible explanations accompanying the outcomes of deep learning models. The integration of interpretability serves a dual purpose: enhancing the grasp of deep learning results and bolstering the models' dependability. With explanatory insights, medical practitioners can readily identify any weaknesses or limitations within the model's predictions, thereby harnessing the model's capabilities more effectively in their clinical practices. Illustrated in Figure 1, our proposed cluster methods not only assign images to distinct clusters but also furnish visual and textual explanations for each cluster.

Figure 1: **Left**: Given input images, we can use a clustering algorithm to label each image into a cluster. But without ground truth, the correctness of the results is unconvincing, and difficult to improve the algorithm. **Right**: With explanation, the result is easy to understand.

To aptly categorize the behavior of each individual microwell image into relevant clusters, our methodology employs a U-net encoder for information extraction from each microwell image. Additionally, we leverage various human-designed features, carefully curated to impart human-derived insights, thereby aiding the model in capturing nuanced aspects of the input microwell images. A multi-head attention layer is harnessed to combine temporal information with human-derived insights upon obtaining the U-net-extracted and human-designed features. By adopting this architectural framework, our model becomes proficient in extracting features that encapsulate the intrinsic attributes of individual microwell images and incorporate temporal dynamics and human-derived contextual knowledge.

After extracting the pertinent features, the Affinity Propagation algorithm (Frey & Dueck (2007)) was chosen as the clustering method. The rationale behind opting for the Affinity Propagation method lies in its inherent ability to circumvent the need for pre-defining an optimal cluster count, aligning well with our problem context. However, owing to the absence of ground truth-remove, a direct assessment of the quality of our clustering outcomes remains elusive. Consequently, to validate both the efficacy of our proposed architectural approach and the clustering algorithm, we introduce two distinct explanation modules tailored to elaborate on our clustering results.

Our explanation module comprises two key components: a visual explanation module and a text explanation module-remove sentence. The explanation module provides insights into our model's focus areas. Achieving this entails employing an attention map to highlight regions of interest as identified by the model. On the other hand, the text explanation module offers comprehensive information about a specific cluster's characteristics, highlighting differentiating factors between this particular cluster and others. Through the combining of these modules, users can readily decipher the rationale behind our clustering methodology, grasp the reasons for the assignment of images to each cluster, and understand the distinguishing traits exhibited by various clusters.

We undertake a series of comprehensive experiments to establish the efficacy of our proposed architecture and the accompanying explanation module. Moreover, we present intra and inter-covariance matrices as additional evidence substantiating the effectiveness of our cluster algorithm. Visualizations of the results serve to reinforce the prowess of our explanation modules visually.

In brief, our contributions encompass several pivotal facets:

- The introduction of a novel dataset, coupled with the proposal of an architectural framework adept at extracting vital information from input microwell images. Notably, a multi-head attention layer is strategically incorporated to synergize U-net-extracted features and human-designed features.

- The formulation of two distinct explanation modules, each playing a crucial role in facilitating a deeper understanding of our model's cluster results. The visual explanation module serves as a guide to discern where the model directs its focus, while the text explanation module illuminates the unique characteristics of clusters and distinguishes them from their counterparts.

- The execution of an extensive array of experiments is meticulously designed to showcase the efficiency of our proposed and explanation modules. The validation process extends to the incorporation of intra and inter-covariance matrices, providing a comprehensive evaluation of the cluster algorithm's effectiveness.

## 2 RELATED WORKS

### 2.1 MEDICAL IMAGE CLASSIFICATION

Since the development of deep learning, more and more people are trying to use deep learning-based methods to help them solve medical problems, including medical image classification. U-Net (Ronneberger et al. (2015)) proposed by Olaf, which is based on FCN (Long et al. (2015)), is a widely-used architecture in medical imaging. The use of the skip connection makes full use of the features of the encoder. Recently, more and more people have focused on using self-supervised or semi-supervised methods to advance the performance of medical image classification. For example, Azizi et al. (2021) noticed that a big self-supervised model can significantly improve the performance of medical image classification. Liu et al. (2020) finds that introducing sample relation consistency can help better extract information from the unlabeled medical data. Zhou et al. (2022) tries to use self pre-training based on Mask Autoencoder (He et al. (2022)) to improve the performance of medical image classification. This paper focuses on explaining the classification without any ground truth.

### 2.2 IMAGE CLUSTERING

Image clustering is an unsupervised machine-learning technique that focuses on finding similar features between different samples and labeling each sample into a cluster. The most typical image clustering algorithm is K-means (Steinhaus et al. (1956)), which was proposed in 1956. Recently, with computer vision and data mining development, more and more clustering algorithms have been proposed. For example, Li et al. (2021) tries to use Contrastive learning to perform image clustering and achieve reliable outcomes on multiple datasets. In another example, Shakeel et al. (2019) introduces a method to improve the performance of clusters on lung cancer detection. In this paper, we use an encoder and a multi-head attention module to extract good image features and improve the medical image clustering performance.

### 2.3 EXPLAINABLE DEEP LEARNING

With the development of deep learning, people are not satisfied with using neural networks without understanding why a decision is made. Hence, there are some researchers who focus on explaining the output of neural networks. There are multiple ways to explain the output of neural networks. One is to generate an attention map based on the input image. Class activation mapping introduced by Bolei Zhou, people have generated better attention maps like Grad-CAM (Selvaraju et al. (2017)), ScoreCAM (Wang et al. (2020a)), RelevanceCAM (Lee et al. (2021)), PullzeCAM (Jo & Yu (2021)). Another is the concept-based method, which was introduced by TCAV (Kim et al. (2018)). Ghorbani et al. (2019) tries to find the explanation based on the concept automatically. Ge et al. (2021) build a GNN to understand the network's decision reasoning better. In this paper, we try to provide a visual explanation and a text explanation about why the network makes a particular decision.

## 3 METHODS

Our proposed methods consisted of four different parts: a feature extractor to extract the features for every group of images, a cluster module to label each group of images to a cluster, a visual explanation module that uses Grad-Cam to provide a visual explanation and a text explanation module which uses the features and templates made by the large language model to generate text explanation. Fig.2 shows the whole architecture of the proposed methods.

### 3.1 FEATURE EXTRACTOR

Our comprehensive feature extraction framework comprises several key components that collectively contribute to the nuanced analysis of cellular imaging. At the heart of this system lies a Convolutional Neural Network (CNN) encoder, meticulously designed to extract intricate information embedded within each individual image. Working in tandem with this encoder is a multi-head attention module, strategically integrated to elucidate temporal intricacies present within groups of

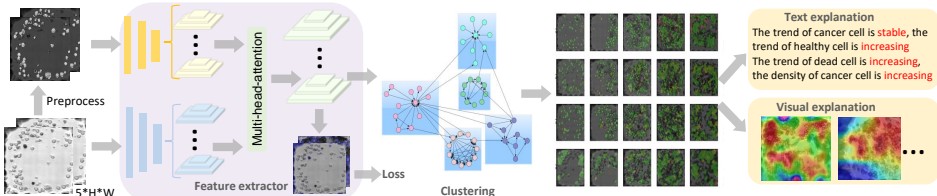

Figure 2: Given input images, our model will first use a feature extractor to aggregate information from the input images, then we will use a clustering algorithm to label each image into a cluster. After getting multiple clusters, we will use text explanation and visual explanation methods to explain why we cluster each image.

images. Additionally, a human-designed feature extractor is employed to distill essential low-level attributes from each input image, providing a holistic representation of the underlying data.

A comprehensive illustration of the architectural intricacies of our feature extraction system can be seen in Figure 2. In our feature extraction module, we rely not only on provided annotations but also on the integration of pseudo-labels. These pseudo-labels are derived from analyzing live and dead cells within an image. This innovative approach enables our system to function effectively even without a complete ground truth.

The generation of these pseudo-labels hinges on a crucial insight: cellular regions exhibit distinct brightness characteristics when contrasted with the background. To harness this distinction, we converted images into the HSV color space, thereby allowing for the precise extraction of foreground and background regions. This process lays the foundation for pseudo-label generation, further enhancing the richness of our input data.

Furthermore, we used several different human-design features. This included using the density of the cancer cell and T cell and the density and rate of change for these cells as human-design features. This can provide statistical information that is typically difficult to capture through neural networks.

The final part of our feature extractor is a multi-head attention module, carefully engineered to integrate insights derived from different images. This module can capture the temporal patterns inherent within image groups, enabling a comprehensive understanding of temporal dynamics and aggregate the human-designed feature.

Using all of these parts, our model can capture multiple pieces of information from the original input and generate features for the cluster module.

## 3.2 CLUSTER MODULE

Upon successful training of the feature extractor, the remaining challenge is the assignment of labels to groups of images, thereby organizing them into coherent clusters. However, a notable hurdle arises from the absence of definitive ground truth regarding which image groups should belong to the same cluster. This complication precludes the direct training of a model for group clustering. Thus, a search for a suitable clustering method ensues, one that can adeptly assign inputs to multiple clusters while accommodating an unknown cluster count.

We select the AffinityPropagation algorithm (Frey & Dueck (2007)) as our clustering approach. Noteworthy advantages of AffinityPropagation include its ability to circumvent the requirement for a predetermined cluster count and its resilience to the influence of initial values, a distinct contrast to algorithms like K-means. However, a significant consideration emerges due to the nature of our feature extractor's output – a high-dimensional feature space rife with dimensions that may prove extraneous for clustering purposes. To address this, a preliminary step involves dimensionality reduction to eliminate redundant dimensions. To this end, we employ T-SNE (t-Distributed Stochastic Neighbor Embedding) (Van der Maaten & Hinton (2008)), a technique that facilitates the transformation of complex high-dimensional data into lower-dimensional data that is conducive to subsequent clustering.

Having reduced the dimensionality of the data, the application of the AffinityPropagation algorithm becomes more feasible. The resultant application of AffinityPropagation yields multiple clusters, each containing a grouping of image sets. However, due to the absence of ground truth, assessing the performance and quality of these clusters is not a straightforward task.

To surmount this challenge and provide insights into the efficacy of the cluster results, two distinct explanation modules were developed. These modules are specifically designed to elucidate the clustering outcomes' rationale by enhancing the cluster assignments' interpretability. To do this, we present visual and text explanation modules. The former serves as a mechanism to highlight the key focus areas within each cluster, employing attention maps to emphasize regions of significance. Meanwhile, the text explanation module articulates each cluster's defining characteristics and elucidates the distinctions between clusters. By seamlessly integrating these two explanation modules, we augment the understanding of our clustering methodology and address the challenge posed by the absence of ground truth.

## 3.3 Visual Explanation Module

In this section, we delve into the methodology behind our generation of visual explanations for each cluster. The objective of our visual explanation module is to discern the focal points of our model's attention within each image, subsequently aiding user understanding of the appropriateness of our model's cluster assignments based on the input images. Taking inspiration from Grad-Cam (Selvaraju et al. (2017)), we leverage gradient information to create attention maps.

However, a noteworthy challenge emerges: our model lacks individual labels for each image, which complicates the direct utilization of gradients derived from target concepts. To circumvent this issue, we adopt a strategic approach post-clustering. Specifically, we assign the cluster index to each image, effectively creating pseudo-labels. Subsequently, we retrain a classifier solely based on the extracted features. Notably, during this classifier training phase, we exclusively update the classifier while keeping the feature extractor unchanged. This deliberate decision maintains the feature extractor's consistency and integrity, safeguarding it against potential distortion by potentially erroneous pseudo-labels.

Following the training of the classifier, we emulate the Grad-Cam methodology by employing a similar configuration. In this context, the target cluster index is employed as the target concept, and the attention map is computed based on the original images. Mathematically, this process can be formalized as follows:

$$
\alpha_k^c = \frac{1}{Z} \sum_i \sum_j \frac{\partial y_c}{\partial A_{ij}^k}
$$
$$
L_{GradCAM}^c = RELU(\sum_k \alpha_k^c A^k)
$$

(1)

Here, $\alpha_k^c$ represents the importance weight attributed to each feature map $A_k$. The significance of these feature maps is aggregated through summation, followed by filtration of extraneous information using a rectified linear unit ($RELU$) function. This process culminates in the creation of an attention map that highlights the pertinent regions within the original images, effectively elucidating the visual cues that our model utilizes for its cluster assignments.

## 3.4 Text Explanation Module

Within this section, our focus shifts towards delineating the process by which we generate text explanations tailored to each cluster. The underlying goal is to provide user-readable explanations that facilitate a clear understanding of the rationale behind our model's cluster assignments.

To achieve this, we initiate the process by constructing a template that serves as a structured framework for generating text explanations. This template contains multiple dimensions of information, encompassing key aspects such as cell densities, proliferation trends, and comparative analysis with other clusters. With this template in place, the subsequent challenge involves populating it with relevant information.

In an effort to bolster the robustness of our explanatory model, we take a novel approach: sidestepping the direct utilization of cancer and T Cell annotations. Instead, we employ these annotations to

train an auxiliary model that predicts cell density based on input images. Leveraging the predictions from this auxiliary model, we gain the ability to generate explanations even in the absence of explicit annotations. This strategic move enhances our approach's adaptability and contributes to the overall resilience of our model's performance.

Akin to the approach taken for generating density-based explanations, the generation of explanations pertaining to proliferation trends follows a similar strategy. Here, a model is trained to predict proliferation trends, thus enabling the prediction of trends-related information for explanation.

A distinguishing feature of our text explanations lies in their comparative analysis. We adopt a comparative perspective to highlight each cluster's unique attributes. Specifically, we meticulously assess the cluster that exhibits the most distinct characteristics compared to other clusters. This distinctiveness forms the explanatory text's crux, emphasizing the distinguishing traits that underlie each cluster's composition.

The resultant text explanations, enriched by these comprehensive elements, allow users to discern the disparities and nuances between clusters readily. Through this mechanism, our methodology transcends the realm of statistical clustering, fostering an in-depth understanding of the intrinsic features that set each cluster apart.

## 4 EXPERIMENTS

### 4.1 EXPERIMENTS SETUP

#### 4.1.1 MICROWELL PRINTING

Approximately 2500 microwells were printed within each well's central 20mm glass bottom area within a 6 well glass-bottom microplate. Microwell printing gel (Enrich Biosystems) was added to each well and placed in the TROVO instrument (Enrich Biosystems) for printing. After a 1hr automated printing process, 2 mL of 1X PBS was added to the wells and incubated at 37°C for 5 minutes to wash. This washing step was repeated three times, and then a complete medium was added for an overnight incubation at 37°C.

#### 4.1.2 CO-CULTURE SETUP

Cancer cells and T cells were stained and combined at a previously optimized ratio in an Eppendorf tube at a total volume of 200 uL. Media was removed from the wells, and the cell mixture was added dropwise throughout the 20 mm microwell. The plate was incubated for 15 minutes at 37°C to allow the cells to settle. Once the incubation was completed, 1.5 mL of complete medium was added to the dish and incubated at 37°C for 1 hr. Following the incubation, the plate was imaged using the Trovo system. Cells were stained within the wells every time prior to imaging. The microwell image dataset was analyzed as described below.

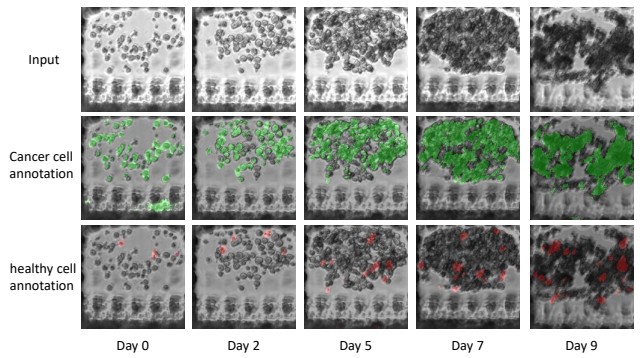

Figure 3: Overview of our dataset. Our dataset contains 2458 groups of images. Every group consists of 5 images collected on different days. For every input image, we provide two different annotations: one is the cancer cells, and the other is the healthy cells.

### 4.2 DATA DESCRIPTION

Within this article, our novel approach is underscored by both an innovative dataset and a unique experimental context. Thus, this section introduces our dataset, highlighting the distinctive aspects that set it apart. Illustrated in

Figure 3, the dataset shows the growth trends of both cancer cells and T cells, serving as a critical foundation for our research.

Each grouping within our dataset comprises a collection of five diverse input images. These images correspond to microwells imaged across different days, offering a temporal perspective on cellular dynamics. We provide two distinct annotations for each input image: one designates cancer cells, color-coded in green, while the other denotes T cells, indicated in red. Our approach extends beyond mere observation as we leverage different conditions to elicit varied responses within each image group.

Our overarching objective revolves around presenting a substantial array of microwell images, totaling 2458 groups. This wealth of data enables us to meticulously track intrinsic features across various groups, subsequently facilitating the assignment of each image group into a relevant cluster. Notably, all images within the dataset maintain uniform dimensions, spanning $173 \times 173$ pixels.

### 4.2.1 IMPLEMENT DETAILS

We implemented our methods using Pytorch and trained them on a single V100-sxm2. During our experiments, we trained our model for 100 iterations with a batch size of 16. The optimizer that we used is SGD, with an initial learning rate of $1e - 4$. The weight decay parameter is set to $1e - 4$

Table 1: Quantitative results of different architecture, the best results are highlighted in **bold**.

| Methods | Acc | SE | SP | PC | JS | DC |
|---|---|---|---|---|---|---|
| Resnet18 (He et al. (2016)) | 93.52 | 86.73 | 96.54 | 94.27 | 87.24 | 91.82 |
| Resnet34 (He et al. (2016)) | 94.05 | 87.26 | 97.15 | 96.18 | 88. 82 | 92.63 |
| HRNet (Wang et al. (2020b)) | 95.39 | 89.47 | 97.18 | 97.49 | 88.67 | 94.18 |
| U-net (Ronneberger et al. (2015)) | 94.25 | 88.15 | 97.02 | 95.14 | 89.21 | 92.10 |
| Ours | **97.84** | **91.45** | **99.03** | **98.87** | **90.56** | **95.06** |

## 5 COMPARISON WITH OTHER MODELS

Given the relative novelty of this subject, a direct comparison between our approach and existing methods is not readily feasible. To provide tangible quantitative insights underscoring the effectiveness of our innovative architecture, we undertake a systematic approach. This involves adapting several alternative architectures to align with our model's framework, thereby enabling a comparative analysis.

The tabulated outcomes, as presented in Table 1, provide the results obtained from alternative methods alongside those yielded by our proposed approach. Upon examination, it becomes evident that our proposed methodologies outperform their counterparts, consistently demonstrating superior performance. This comparative evaluation substantiates our claim of attaining the most effective outcomes within the context of the presented problem domain.

## 6 ABLATION STUDY

### 6.1 EFFECTIVENESS OF THE PREPROCESSING MODULE AND TEMPORAL FEATURES

This section is dedicated to performing an ablation study, wherein we aim to showcase the effectiveness of both our preprocessing module and the integration of temporal features. As shown in Table 2, it becomes evident that introducing preprocessing and temporal features yields a discernible enhancement in our model's performance. This augmentation is attributed to the distinct advantages offered by these two components.

The preprocessing phase is particularly effective and impactful in enabling our model to refine its focus on foreground elements, consequently contributing to a more accurate analysis of relevant aspects. In parallel, the incorporation of temporal features amplifies the informational richness compared to individual images. The comprehensive analysis provided by these temporal features significantly contributes to the model's performance, as corroborated by the results presented.

Table 2: Ablation study for the existence of pre-processing and temporal features, the best results are highlighted in **bold**.

| Methods Preprocessing | Temporal Feature | Acc | SE | SP | PC | JS | DC |
|---|---|---|---|---|---|---|---|
| ✗ | ✗ | 96.35 | 89.24 | 97.97 | 96.66 | 87.45 | 93.19 |
| ✓ | ✗ | 96.52 | 89.73 | 98.54 | 97.27 | 88.24 | 92.82 |
| ✗ | ✓ | 97.42 | 91.09 | 98.86 | 98.79 | 89.36 | 94.16 |
| ✓ | ✓ | **97.84** | **91.45** | **99.03** | **98.87** | **90.56** | **95.06** |

Table 3: Ablation study for the existence of human-designed features, the best results are highlighted in **bold**.

| Methods Density | Trend | Acc | SE | SP | PC | JS | DC |
|---|---|---|---|---|---|---|---|
| ✗ | ✗ | 94.25 | 86.97 | 94.26 | 93.11 | 83.20 | 90.01 |
| ✓ | ✗ | 95.12 | 87.73 | 95.18 | 96.04 | 85.28 | 91.42 |
| ✗ | ✓ | 96.26 | 90.83 | 95.07 | 97.19 | 87.16 | 93.16 |
| ✓ | ✓ | **97.84** | **91.45** | **99.03** | **98.87** | **90.56** | **95.06** |

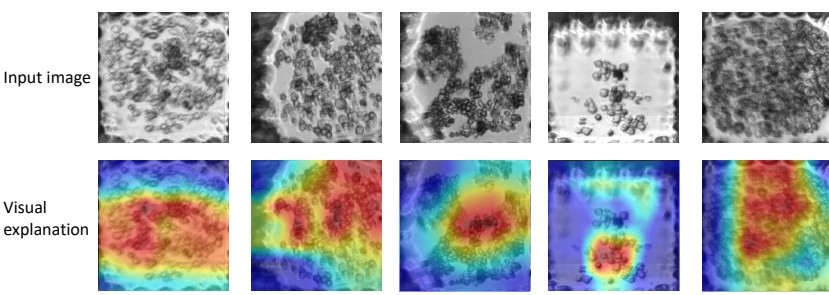

Figure 4: Our methods can highlight the region where the model focuses on and visualize them using an attention map

## 6.2 EFFECTIVENESS OF HUMAN-DESIGNED FEATURES

Within this section, we conducted an ablation study to substantiate the efficacy of our human-designed features. The results, as highlighted in Table 3, illustrate that the incorporation of human-designed features yields a discernible improvement in our model's performance. Delving deeper, the rationale behind this enhancement lies in the unique attributes of these human-designed features. Their ability to contribute statistical features that are intricate and not readily learnable by neural networks imparts a decisive advantage, thereby bolstering our model's overall performance.

## 7 VISUALIZATION OF OUR VISUAL EXPLANATION MODULE

This section shows examples of the visual explanation results in Fig.4. As seen in Fig.4, the model highlights the region of the image where the model focuses. Based on the fact that the highlighted region matches the cells, it proves that our model attempts to capture the information from the cell.

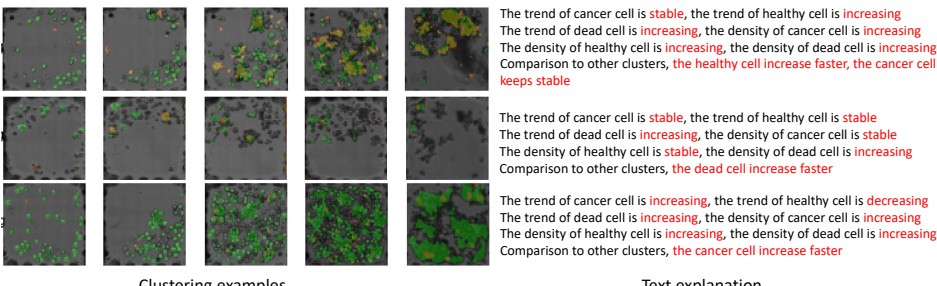

Figure 5: our methods can provide the explanation for each cluster and a comparison between this cluster and other clusters. Here, we randomly pick one group of images for every cluster.

## 8 RESULTS OF OUR TEXT EXPLANATION MODULE

In this section, we show the result of our text explanation module. As shown in Fig.5, our methods can not only provide the explanation for each cluster but also provide a comparison between this cluster and other clusters.

## 9 INTRA AND INTER COVARIANCE MATRIX

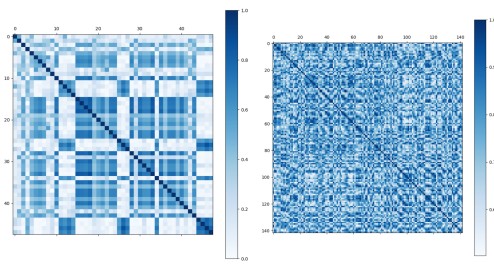

Figure 6: We visualize intra and inter-covariance matrices for different clusters. The left image is the inter-covariance matrix, with low similarity between different clusters. The right image is the intra-covariance for a random cluster. Notice that the start point for the right image is 0.6, meaning that for a specific cluster, every image in this cluster has high similarity.

In order to provide a more comprehensive illustration of the validity of our cluster algorithm, we present the intra and inter-covariance matrices in this section, as depicted in Figure 6. The intent behind showcasing these matrices is to elucidate the effectiveness of our clustering methodology through visual representation.

For the calculation of the intra-covariance matrix, we adopt a methodical approach. Commencing with the random selection of a cluster, we proceed to compute the cosine similarity between the feature vectors of every pair of images within that cluster. The outcomes are then presented in the form of a matrix, effectively encapsulating the cosine similarity between various image pairs. A notable trend emerges, as observed in the right image of Figure 6: images within the same cluster exhibit significantly high cosine similarity, a testament to the accuracy of our clustering approach.

The process for generating the inter-covariance matrix follows a similar methodology. The starting point involves the calculation of cluster centers for each individual cluster. Subsequently, the cosine similarity between these cluster centers is determined. As depicted in the left image of Figure 6, different clusters manifest relatively low cosine similarity, reaffirming the distinctiveness of our clustering outcomes.

The visual representation of these covariance matrices provides a compelling visualization of the efficacy of our cluster algorithm. Inter-cluster similarity remains low after clustering, while intra-cluster similarity remains notably high. This observation underscores the robustness of our clustering approach in discerning distinctive patterns and grouping images effectively.

## 10 CONCLUSION

In summary, we have introduced an innovative approach that acquires information from individual images and captures cellular dynamics. Following the extraction of features, our method employs a clustering algorithm to categorize each image group into distinct clusters. The absence of ground truth poses a challenge in evaluating our clustering algorithm's performance quality.

To address this predicament, we have integrated two distinct explanation modules. By leveraging textual and visual explanations, we have devised a comprehensive means of comprehending the underlying rationale behind the model's predictions. This multifaceted approach enables us to gain valuable insights into the decision-making process.

We believe that the versatility of our proposed methodology extends beyond the confines of the current domain. With its potential to effectively tackle various challenges, we anticipate that our approach will find application in diverse areas while delivering quality performance.

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
