# Supplement of Explainable medical image clustering

## 1 TROVO SYSTEM

The Trovo system leverages the benefits of conventional petri dish or microtiter plate bulk cultures and transforms these platforms into thousands of microchambers, enabling easy operation, compatibility with existing protocols, and optimal cell health. The hydrogel-based well system segregates cells using hydrogel lithography while maintaining rigidity, transparency, and customizability. Our microwell generation software is fully parameterized, producing 300-micron-high walls and 2.5k microwells in 10 minutes. The hydrogel microwell generation device, which can be seen in the appendix Fig.1 uses hydrogel lithography, allowing customizable microwell size and shape. Essentially, a 405nm laser induces the gelation of various photopolymerization hydrogels, which are enzymatically digestible. The optimal composition of hydrogels forms optically transparent grids that bind well to standard culture dishes. For suspension cultures, cells at the bottom of microwells are unaffected by macroscopic disturbances such as medium changes. This micro-trapping property ensures high-density clone segregation and the registration of individual clone behavior. all while providing excellent cell restraining capacity.

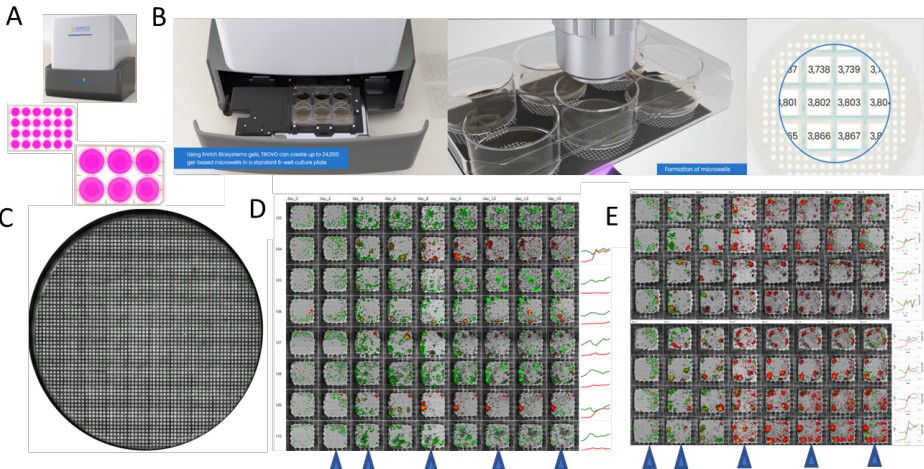

Figure 1: Screening long-term tumor killing and T cell proliferation/persistence behavior of CAR-T clones using Trovo. (A) The hydrogel microwell system can be made on a 6-well or 24-well culture dish using the Trovo system. (B) T-cell tumor co-cultures are seeded into each microwell, observed daily, analyzed, and isolated based on the behavior of each microwell. (C) A "bird's-eye" view of a typical microcompartment co-culture on day 8. (D, E) Collection of microwell images over several weeks showing distinct clonal behavior. The time series data of microchambers were typically collected from 2300 chamber experiments on Trovo. (Yellow/Red: T cells, Green: live tumor, Grey: dead tumor).

## 2 T-SNE VISUALIZATION

This section entails the visualization of our clustering outcomes through the utilization of T-SNE, as depicted in Figure 2. A discernible pattern emerges from this visualization: each cluster is distinctly

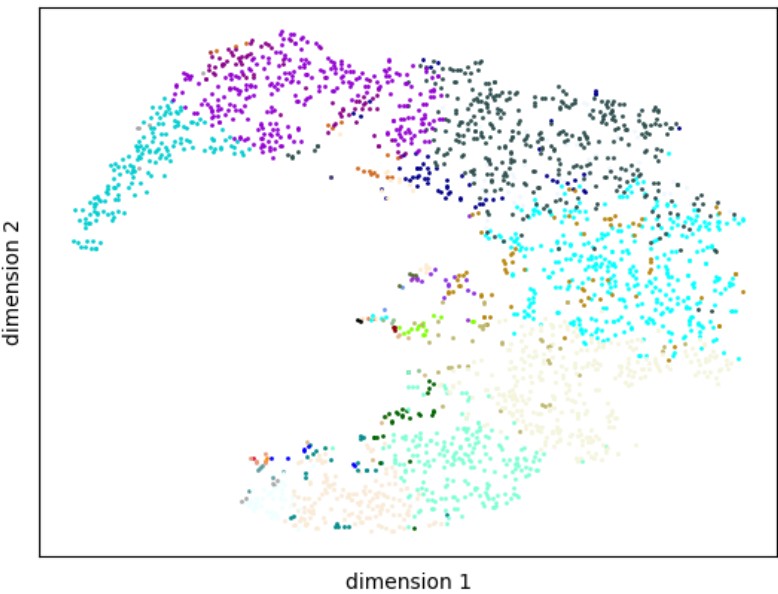

Figure 2: Result of T-SNE after clustering. Each cluster is separated which proves the effectiveness of our proposed method.

segregated within the T-SNE plot. Notably, our approach showcases the capability to accurately predict clusters even when the corresponding image count is limited.

Upon closer examination of these particular instances, we recognize that the images within these clusters possess unique features that set them apart from the other clusters. This realization underscores the efficacy of our methodology in identifying latent characteristics that facilitate accurate clustering, even in scenarios where the available image count is sparse.