# OpenReview forum: "Explainable medical image clustering"
_ICLR.cc/2024/Conference — Submitted to ICLR 2024_

### Official Review · Reviewer_ig3J · 2023-10-30

**Soundness:** 2 fair
**Presentation:** 2 fair
**Contribution:** 2 fair
**Rating:** 1
**Confidence:** 4

**Summary:**

This paper studies medical microwell images containing cancerous and healthy cells, which are annotated in the provided data set.

The goal of the paper is to cluster the images in an informative way, in the absence of ground-truth labels for clustering, and provide a human-interpretable explanation for the clustering.

The proposed method can be separated into four modules: first, a feature extractor is trained, which consists of a u-net architecture combined with a multi-head attention module.  The images are then clustered based on the extracted features using affinity propagation.  A visual explanation of the clusters is given by adopting grad-cam and taking the cluster labels as the target concepts.  A text explanation is provided based on auxiliary predictions of cell density and proliferation.

**Strengths:**

This paper examines an interesting and important question, with relevance to cancer/immunotherapy research.  In this context, the idea of studying the problem with an emphasis on explainable learning is well-motivated, in order to gain more insight on what is being learned by the network and to have more confidence in the predictions made by the network.

**Weaknesses:**

weaknesses can be grouped into four categories as follows, in order of importance:

1. novelty

Each module in the proposed framework is largely leveraging existing work, with basic adaptation to the studied image data.  Feature extraction seems to be u-net with multi-head attention and some hand-designed features.  Clustering is T-SNE followed by affinity propagation.  Visual explanation is GradCam using assigned cluster labels as target concepts.  Text explanation is looking at the output of auxiliary image classification models.

2. clarity of details

Important details in the paper are not clearly conveyed and/or missing.  As examples: The description of the feature extractor is unclear in multiple respects - what are the specifics of the architecture, what are the specifics of the pseudo-labels, what are the specifics of the hand-designed features and how are these features integrated, what is the loss function, what is the training data.  What is the dimensionality of the feature extractor output?  How are the cancer and healthy cell annotations obtained, are they dense (complete) labelings, are they noisy?  In the result tables, the headings are not explained - SE, SP, PC, JS, DC.

3. experimental validation

Very limited quantitative evaluation and comparisons against existing baselines are given, particularly for the main goal of the paper which is the clustering of the images.  As far as qualitative evaluation, there are not concrete examples of interesting findings or observations that can be derived from the final clustering.

4. clarity of writing

I would encourage the authors to focus on clear, simple writing, with the goal of communicating information in a clear and concise manner.  As a practical specific example, many adjectives and adverbs can be dropped.  For instance, "comprehensive" is used 11 times in the paper, "meticulously" is used 4 times in the paper; simply dropping these terms has no impact on the meaning but makes the writing more concise.

**Questions:**

See weaknesses above.

---

### Official Review · Reviewer_5T7W · 2023-10-31

**Soundness:** 1 poor
**Presentation:** 1 poor
**Contribution:** 2 fair
**Rating:** 3
**Confidence:** 5

**Summary:**

The authors propose a cell dataset for cancer and healthy cells. The paper follows a sequential process of feature extraction, clustering, and visual and textual explanations of the clustering algorithm. Authors also provide ablation study of different modules of the proposed method and evaluate on the proposed dataset.

**Strengths:**

Authors provide a new dataset for cancer and healthy cells. However, they do not discuss the availability of dataset to use for research community. Authors use clustreing to overcome the issue of ground truth and use explanation modules to get the validity of the generated clusters. Authors also give a detailed ablation and evaluation.

**Weaknesses:**

The paper is not easy to follow. There are many parts missing which makes the paper hard to understand fully. My detailed comments are as follows.

1> The paper proposes a new cell dataset, however, it does not discuss the drawbacks of previous datasets. Authors should provide a comparative study and explain why a new dataset is necessary. It is not clear if annotations are available for the dataset. If yes, how annotations are prepared should be mentioned. In Fig. 3 cancer cell and healthy cell annotations are given, however, they overlap. For example, in ‘Day0’ image, the same cell is marked as both healthy and cancer cell.
2> Fig.2 is not clearly explained. Forward pass and backpropagation is not clear. Also, loss function is not defined.
3> ‘Preprocess’ mentioned in Fig. 2 is not explained in the paper.
4> Network architecture and training are not clearly mentioned.
5> How are human-designed features computed and why can't neural networks capture these features?
6> Are explanations generated using Grad-Cam and text modules verified by humans?

**Questions:**

please see the weakness section.

---

### Official Review · Reviewer_peGV · 2023-11-02

**Soundness:** 1 poor
**Presentation:** 1 poor
**Contribution:** 1 poor
**Rating:** 3
**Confidence:** 5

**Summary:**

The paper presents a temporal dataset of microscopy cell images. Each set of images is taken over 5 day period and contains annotations for healthy and cancerous cells under different medication experiments. The authors propose a deep learning model that consists of a U-Net encoder (feature extractor) in addition to manual features representing density and rate of change in cancer and T-cells. These features are extracted for each image and passed to multi-head attention to capture attention over the temporal data. The resulting features are used to perform clustering. To explain the resulting clusters, Grad-cam is used to visualize important parts of the images. Text explanations based on transforming manual features into text are also proposed.

**Strengths:**

- The author create a public temporal microscopy dataset with cancer and healthy cells annotations.
- The paper proposes a pipeline for processing the temporal data.

**Weaknesses:**

- The method is not clear. For instance:
  - It is not clear how the pseudo labels for live/dead cells are derived and how they are used. Later there is only mention of using HSV to separate foreground and background.
  - How is the feature extractor trained on and what is the loss function?

- The text explanation is based on manual features such as density where a model is trained to predict them. They don't have anything to do with the features used in the clustering. So they really don't explain the clustering.

- In Fig 3, the annotations for cancer and healthy cells overlap. This is especially obvious in the day 9 images.

- In Table 1, the authors compare unet encoder with other architectures including resnet. Without the skip connections, a UNet encoder is basically a large CNN.

- The authors present the dataset as temporal sequences based on variation in treatment. However, the results do not show the clustering correlate with any outcome or treatment.


- Minor:  In page 2:
  - Typo in the sentence "However, owing to the absence of ground truth-remove"
  - Typo in the sentence "a text explanation module-remove sentence."

**Questions:**

- In Fig 2 caption: "then we will use a clustering algorithm to label each image into a cluster." Does that mean each individual image is clustered? I thought it was each temporal sequence.
- In the section 3.2 Cluster Module, does group refer to a cluster or a temporal sequence?
- The scores presented in Table 1, what do the abbreviations mean? and what is the task?

---

### Official Review · Reviewer_ENtk · 2023-11-03

**Soundness:** 2 fair
**Presentation:** 2 fair
**Contribution:** 2 fair
**Rating:** 3
**Confidence:** 3

**Summary:**

The authors introduce a cell dataset that captures the developmental trend of cancer cells and T Cells under the influence of diverse experimental conditions . The authors also present an approach to both cluster input images and elucidate the rationale behind their grouping. They leverage a U-net encoder for individual microwell image information encoding and multihead attention for information encapsulation across different time points

**Strengths:**

The introduction of a novel dataset with unique acquisition protocols relative to what is encountered usually in the community, coupled with the proposal of an architectural framework adept at extracting vital information from input microwell images. A multihead attention layer is incorporated to rationalise U-net-extracted features and human-designed features. The experimental design and the engineering of different off-the-shelf components to work on microwell data, while not novel from a methodological lens, is an appreciable contribution to the ML for healthcare community.

**Weaknesses:**

There is very little offered in terms of methodological novelty. The paper is a collection of existing (and widely used) ideas in literature glued together in a manner to address the proposed application. As such, this would be a nice contribution in a more focused venue that brings together practitioners in the domain addressed by the paper (and the introduction of the dataset would indeed be a great addition).

The writing and elucidation of ideas needs reworking. Several sections are not clearly written. One issue that crops up is whether there is sufficient clarity with regards to the biological concepts described in relation to the dataset proposed, assuming the typical ICLR audience. The machine learning methods used can be described more clearly. Clarity of the images used needs improvement as well -  this is important since a new dataset rich in imaging features and artifacts is being introduced as a key contribution.

The ablation studies need to be more detailed and described better. Currently, the section is written in a manner that is not particularly informative or helpful in discerning the key messages of the paper. Please consider using Supplementary materials to address content that doesn't fit within page limits.

Typo: 4.2.1 'IMPLEMENT' DETAILS --> IMPLEMENTATION DETAILS

**Questions:**

Please describe the actual methodological novelties pursued in this problem?

---

### Meta-Review · Area_Chair_e7ym · 2023-12-08

**Metareview:**

Based on the submission and reviews, the main points that have been raised are summarised as follows.

Strengths:

1. This work examines an interesting and important question; the idea is well motivated.
2. The introduction of a novel dataset is valuable.
3. The experimental design and the engineering process is an applicable contribution.

Issues:

1. There is very little offered in terms of methodological novelty.
2. The writing and clarity of this work need to be improved.
3. The ablation studies need to be more detailed and described better.
4. Very limited quantitative evaluation and comparisons against existing baselines.

The authors do not provide feedback to address the above raised issues. All the ratings are leaning towards rejecting this work. After reading the submission, I share the similar opinion. This work has its merits in developing a novel dataset and conducting experimental evaluation. However, its technical novelty and contribution are limited and the work shall be better presented. It is hoped that the reviews will help the authors to further strengthen this work.

Regards, AC

**Justification For Why Not Higher Score:**

The authors do not provide feedback to address the above raised issues. All the ratings are leaning towards rejecting this work. After reading the submission, I share the similar opinion. This work has its merits in developing a novel dataset and conducting experimental evaluation. However, its technical novelty and contribution are limited and the work shall be better presented.

**Justification For Why Not Lower Score:**

N/A

---

### Decision · Program_Chairs · 2024-01-16

Reject